# Occurrence of Free Amino Acids in the Source Waters of Zhejiang Province, China, and Their Removal and Transformation in Drinking Water Systems

**Yulong Yang** [1] , **Qi Yu** [2], **Ruonan Zhou** [2], **Jiao Feng** [2], **Kejia Zhang** [1], **Xueyan Li** [3], **Xiaoyan Ma** [2,*] **and Andrea M. Dietrich** [4]

[1] College of Civil Engineering and Architecture, Zhejiang University, Hangzhou 310058, China; yulongy@zju.edu.cn (Y.Y.); zkj1025@163.com (K.Z.)

[2] College of Civil Engineering and Architecture, Zhejiang University of Technology, Hangzhou 310014, China; 18989485209@163.com (Q.Y.); zrn25710@163.com (R.Z.); tangliangjie628@163.com (J.F.)

[3] School of Environmental Science and Engineering, Suzhou University of Science and Technology, Suzhou 215009, China; lxyhit@sina.com

[4] Civil and Environmental Engineering, Virginia Polytechnic Institute and State University, Blacksburg, VA 24061, USA; andread@vt.edu

\* Correspondence: mayaner620@163.com; Tel.: +86-136-3419-0448

**Abstract:** Free amino acids (FAAs) are key components of the global nitrogen cycle and important disinfection byproduct (DBP) precursors. The knowledge gap of FAA occurrence in source and engineered water is discussed in this paper. Solid phase extraction and post column derivatization was combined with gas chromatography–mass spectrometry to simultaneously detect µg/L concentrations of FAAs. This method efficiently detects alanine (Ala), threonine (Thr), serine (Ser), valine (Val), leucine (Leu), isoleucine (Ile), proline (Pro), aspartic (Asp), phenylalanine (Phe), and glutamic acid (Glu) with good linearity, accuracy, and precision. An investigation of FAAs in surface waters in Zhejiang Province found concentrations of 1.48–14.73 µg/L Ala, 0.20–2.39 µg/L Thr, 0.41–7.84 µg/L Val, 0.21–6.86 µg/L Ser, 0.11–4.16 µg/L Leu, 0.57–1.54 µg/L Ile, 0.24–8.06 µg/L Pro, 0.42–4.73 µg/L Asp, 0.30–3.01 µg/L Phe, and 0.12–3.83 µg/L Glu. Phe and tyrosine (Tyr) exhibited higher trichloromethane (TCM) formation (1029–1148 µg/mmolAA) than dichloroacetonitrile (DCAN) formation (333–347 µg/mmolAA). Asp and Glu demonstrated the opposite trend: higher DCAN (570–1106 µg/mmolAA) formation than TCM (137–506 µg/mmolAA).

**Keywords:** solid phase extraction; disinfection byproduct precursor; free amino acids; C-DBP; N-DPB

## 1. Introduction

Globally, water resources, especially surface water receiving wastewater effluent and agricultural runoff, are increasingly polluted with dissolved organic nitrogen (DON) [1]. During the disinfection process of chlorination or chloramination, DON can generate halogenated nitrogenous disinfection byproducts (N-DBPs), such as haloacetamides (HAcAms), halonitromethanes (HNMs), and haloacetonitriles (HANs) [2]. Free amino acids (FAAs) are a typical type of DON, accounting for 15–35% of all DONs in source water [3]. FAAs are highly water soluble and poorly removed during water treatment; thus, they become key precursors that form a variety of DBPs, including trihalomethanes (THMs), haloaceticacids (HAAs), haloacetonitriles (HANs), haloacetamides (HAcAms), and halonitromethanes (HNMs) [4,5]. When chlorinated, the amino acid tyrosine produces 4-chlorophenol, dichloroacetonitrile, dichloroacetamide, chloroform, and dichloroacetic acid; the maximum disinfection byproduct yields occur at pH 7 or 8 [6]. THMs and HAAs are

halogenated carcinogenic DBPs regulated by governments and agencies worldwide, including the U.S. Environmental Protection Agency (USEPA). Compared with regulated DBPs, unregulated DBPs, including nitrogenous DBPs (N-DBPs), may be more carcinogenic and mutagenic [7–10]. A variety of nitrogenous organic compounds, including peptides, amino acids, and humic acids, are important precursors of N-DBPs. Bond et al. demonstrated that phenylacetonitrile results from chlorination of phenylalanine [5]. The N-DBP phenylacetonitrile was present in drinking water, produced by chlorination of surface waters; this N-DBP has an unpleasant odor with an odor threshold concentration of 0.2 ppt-v. In the literature, few articles discuss the detection and occurrence of amino acids, and even fewer report concentrations for individual amino acid species. Brosillon [11] reported that total amino acid (AA) concentration levels ranged from 0 to 19 μg/L in raw water sampled from three drinking water plants in Paris. Berne et al. [12] conducted a survey of several drinking water treatment plants in western France, and found that the concentration of total AAs ranged from 0 to 30 μg/L. In order to measure individual FAAs, Zuo utilized solid phase extraction (SPE) coupled with liquid chromatography–tandem mass spectrometry (LC-MS/MS) and reported mean concentrations of 2.00 μg/L proline (Pro), 5.33 μg/L leucine (Leu), 3.33 μg/L tryptophan (Try), and 5.00 μg/L tyrosine (Tyr) in three different source waters [13]. Isoleucine (Ile), glycine (Gly), alanine (Ala), asparagic acid, and serine (Ser) were dominant amino acids identified by several researchers [12,14]. Of note, monitoring DBP precursors is essential for protecting public health; however, there is little data on the occurrence of FAAs in source waters or treatment systems. There is a lack of data on the content of natural organic compounds in water sources in China, such as amino acids. At present, AAs are the main precursors of nitrogen-containing disinfection byproducts (N-DBPs), so it is necessary to investigate the concentration of AAs in water environments.

Detection technologies for FAAs include capillary electrophoresis, near-infrared spectroscopy, gas chromatography, liquid chromatography, and liquid chromatography–mass spectrometry [15–19]. However, UV/Vis spectrometry cannot identify individual AAs in aqueous mixtures. As liquid chromatographic methods with good sensitivity, accuracy, rapid analysis, and reliability are widely applied to detect FAAs, pre- or post-column derivatization is not essential [20]. However, if the flow pattern of the mobile phase changes, diffusion and retention of any specific separated material will significantly broaden the chromatographic peak and lower column efficiency. Capillary electrophoresis detection will be less sensitive when connected with ultraviolet absorption spectrometry, because of the lesser injection, small capillary diameter, and short optical path. Near-infrared spectroscopy is suitable for the quick and simultaneous detection of amino acids, however, its accuracy and sensitivity need to improve. Considering the limited actual conditions in our lab, detection accuracy, and sensitivity, gas chromatogram combined with mass spectrometry was selected.

The purpose of this work is to determine occurrence of FAAs by using a developed solid phase extraction and derivatization, combined with gas chromatography–mass spectrometry (GC-MS) and DBP formation of typical amino acids that occur in a water source.

## 2. Materials and Methods

### 2.1. Reagents and Instruments

All the FAA standards were obtained from Guangzhou Zuoke Biotechnology Development Co., Ltd. Trifluoroacetic anhydride solution (standard 99%, Meryer, Shanghai, China) was purchased from Shanghai Miner Chemical Technology Co., Ltd. The methanol, ethyl acetate, and dichloromethane were chromatographically pure (Wokai, Beijing, China). The hydrochloric acid, n-butanol, ammonia, and other reagents were of analytical grade, and they were acquired from Sinopharm Chemical Reagent Beijing Co., Ltd. Ultra-pure reagent water (18 MΩ) was prepared by a Millli-Q apparatus (Barnstead Nanopure, Thermo Scientific, Waltham, MA, USA). Hydrochloric acid/n-butanol solution (HCl = 3.5 mol/L) were used for derivation. Strong cation exchange filler (SCX, 3 mL/60 mg) cartridges

(CNW, Dusseldorf, Germany) were used for pretreatment of the amino acids in the water sample, and C18 (CNW, Shanghai) was used for purifying.

A 12-head solid phase extraction device (Visiprer DL, Supleco, Bellefonte, PA, USA) was used to conduct solid phase extraction. A constant temperature nitrogen blowing apparatus (N-EVAP-111, Organomation, Bellefonte, PA, USA) was used for extractant drying. A gas chromatography mass spectrometer (GC/MS-QP2010Plus, Shimadzu, Kyoto, Japan) was used for separation and quantification of the FAAs.

## 2.2. Derivatization Procedure

Solid phase extraction treatment: First, 4.5 mL methanol and 9 mL 0.1 M HCl were used to consecutively rinse the SPE extraction column. Next, a 1 L water sample, adjusted to pH 1.3 by adding concentrated HCl (12 mol/L), was passed through the column at a flow rate of 2 mL/min. The SPE column containing FAAs was eluted with three sequential 3 mL volumes of methanol containing 5% ammonia. The eluent was dried with a stream of nitrogen gas, and re-dissolved in 1 mL 0.1 M HCl.

Derivatization treatment: A volume of 100 μL of FAA stock solution or SPE eluents of source or drinking water samples were transferred into 2 mL glass centrifuge tubes, together with 50 μL of $CH_2Cl_2$. After drying with a stream of nitrogen gas, 50 μL of hydrochloric acid/n-butanol (3.5 mol/L HCl in n-butanol) was added. After being tightly sealed, the tubes were reacted in an oven at 120 °C for 1 h. Another 50 μL $CH_2Cl_2$ was added, and the sample was again dried with nitrogen gas. Subsequently, 100 μL of ethyl acetate and 100 μL of trifluoroacetic anhydride were added, and the glass tubes were heated in the oven at 120 °C for another 30 min. After the reaction, the residue was dried with nitrogen and dissolved in 500 μL of $CH_2Cl_2$. Finally, 1.0 μL of the $CH_2Cl_2$ was injected directly into GC-MS.

The linear regression equation of each amino acid showed a good linear relationship, and the total recovery rate for FAAs was about 80%, which indicated that a gas chromatogram combined with mass spectrum can meet the requirements of FAA detection.

## 2.3. Conditions of GC-MS

The GCMS-QP2010Plus (Shimadzu Corporation) was operated in splitless mode. The carrier gas was pure helium, with flow rate of 1 mL/min. The injector and detector were set at 250 °C and 280 °C, respectively. The 30 m, 0.25 mm id, and 0.25 μm film thickness Rtx-5 column was manufactured by Shimadzu. The temperature program was 40 °C initially for 5 min, then it was increased to 300 °C (held for 2 min) at a rate of 15 °C/min. For the MS detector, the filament current was 220 mA, and the ion source temperature was set at 200 °C. The mass scan range was 40–600 amu.

## 2.4. DBP Formation Potential of FAAs

The chlorination experiments were conducted in a 250 mL glass conical flask with a ground neck, covered by tinfoil to avoid light. The glass conical flask was sealed with cling film. Four model FAAs, including Phe, Tyr, Asp, and Glu, were prepared at an initial concentration of 1 mmol/L. The dosage of chlorine was 10 mmol/L. The mixed FAAs and chlorine were sealed for reaction for 3 days at 25 °C. Each FAA solution had three replicates; one for determination of residual chlorine at different intervals, and the other two to determine the concentration of THMs, HAAs, HANs, and TCNM.

THMs (i.e., $CHCl_3$, $CHCl_2Br$, $CHClBr_2$, and $CHBr_3$) were analyzed according to the standard method. HAAs (i.e., $CH_2ClCOOH$-MCAA, $CHCl_2COOH$-DCAA, and $CCl_3COOH$-TCAA) were determined using the USEPA method 552-3. Haloacetonitriles ($CH_2ClCN$-MCAN, $CHCl_2CN$-DCAN, and $CCl_3CN$-TCAN) were determined using the USEPA method 552-1. Analysis of trihalonitromethane (TCNM) was carried out on a gas chromatography system coupled with an electron capture detector, according to the methods proposed by De Vera [21].

However, 18 kinds of amino acids were selected as targets for derivative-GCMS analysis in this paper. For unknown reasons, only 10 kinds of amino acids were identified, leading to a failure to investigate all kinds of amino acids in the water environment and water treatment technology.

*2.5. Sample Information*

Water resources and the water plant of H city were the main investigation objects; the source water, water plant effluent, and finished water were collected for amino acid detection four times from December 2017 to November 2018. DT River and LHT Reservoir were the resources of CX waterworks, which is a conventional treatment waterworks that undertakes coagulation, sedimentation, filtration, and disinfection treatment. TH waterworks intakes water from T lake and is equipped with advanced treatments (i.e., pre-ozonation, coagulation, sedimentation, sand filtration, ozonation, and granular activated carbon (GAC) filtering and disinfection). The two waterworks both use liquid chlorine disinfection.

FAAs were measured in a series of water samples collected in December 2017 and January, May, and November 2018 from six waterworks (i.e., JX, NXQ, QT, XF, CX, and TH waterworks) that treat surface water. Of note, upstream urban sewage, animal husbandry sewage, and biological metabolism in the river are the main sources of amino acids. A wastewater treatment plant (BL) in Zhejiang Province that collects domestic sewage and applies the oxidation ditch process was also investigated. CX, XF, and JX waterworks apply conventional treatment processes; the advanced treatment processes of oxidation, activated carbon, or membrane filtration are applied at the NXQ, QT, and TH waterworks. The samples were collected in 4 L plastic drums and brought back to the laboratory for analysis as soon as possible. A map with sampling locations is shown in Figure 1, and the water quality of six water plants can be seen in Table 1.

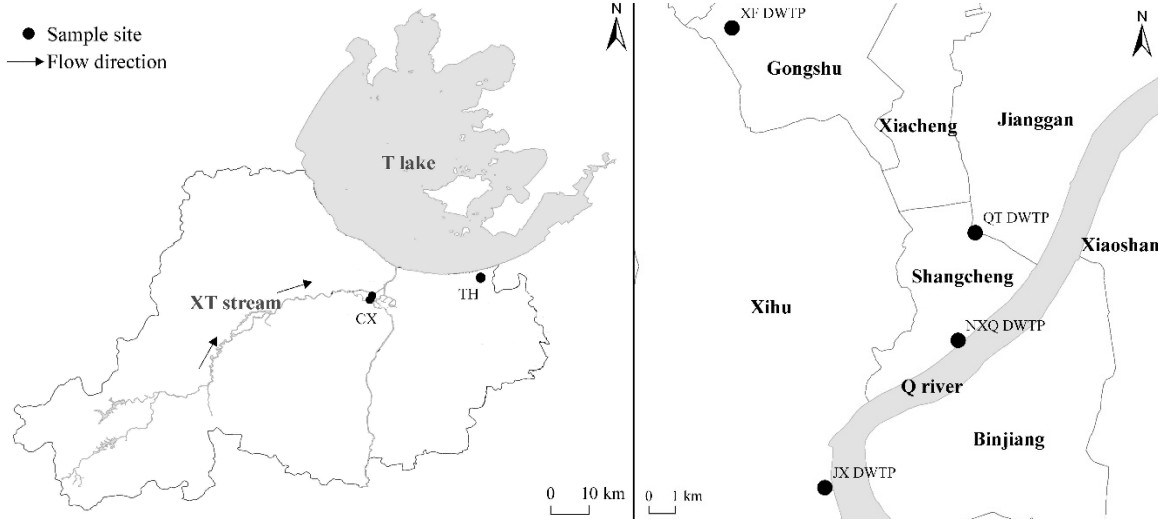

**Figure 1.** Map of the sampling sites. (DWTP = drinking water treatment plant).

**Table 1.** Water quality of sampled waters.

| Water Quality Samples | | TOC mg/L | Conductivity µs/cm | pH |
|:---:|:---:|:---:|:---:|:---:|
| ① | RW | 111.6–121.4 | 1500–1810 | 8.0–8.2 |
|    | FW | 2.3–3.1 | 710–1310 | 6.9–7.9 |
| ② | RW | 1.9–4.2 | 101–120 | 7.4–7.8 |
|    | FW | 1.4–2.5 | 98–118 | 7.2–7.5 |
| ③ | RW | 2.5–3.9 | 88–130 | 7.4–7.9 |
|    | FW | 1.0–2.7 | 80–104 | 7.2–7.5 |
| ④ | RW | 1.9–4.2 | 110–129 | 7.2–7.7 |
|    | FW | 1.2–2.6 | 96–107 | 7.1–7.4 |
| ⑤ | RW | 2.7–4.2 | 117–222 | 7.4–7.8 |
|    | FW | 2.1–3.4 | 121–189 | 7.3–7.8 |
| ⑥ | RW | 2.2–5.9 | 99–336 | 7.1–7.5 |
|    | FW | 1.3–1.6 | 105–110 | 7.0–7.2 |
| ⑦ | RW | 3.8–8.7 | 314–452 | 7.5–7.8 |
|    | FW | 1.6–3.2 | 301–455 | 7.1–7.2 |

①–⑦ represent samples from the water treatment plants of FH, JX, NXQ, QT, XF, CX, and TH; RW and FW mean raw water and finished water, respectively.

## 3. Results and Discussion

As this paper contains many abbreviations, a detailed explanation of abbreviations is shown in Table 2 for the reader's convenience.

**Table 2.** Glossary explaining of acronym.

| Acronym | Glossary Explaining |
|:---:|:---:|
| FAAs | free amino acids |
| DBPs | disinfection by-products |
| N-DBPs | nitrogenous disinfection by-products |
| C-DBPs | carbon disinfection by-products |
| THMs | trihalomethanes |
| HAAs | haloaceticacids |
| HANs | haloacetonitriles |
| HAcAms | haloacetamides |
| HNMs | halonitromethanes |
| TCM | trichloromethane |
| DCAN | dichloroacetonitrile |
| DCAA | dichloroacetic acid |
| TCAA | trichloroacetic acid |
| TCNM | trichloronitromethane |
| Ala | alanine |
| Thr | threonine |
| Ser | serine |
| Val | valine |
| Leu | leucine |
| Ileu | isoleucine |
| Pro | proline |
| Asp | aspartic |
| Phe | phenylalanine |
| Glu | glutamic acid |
| Try | tryptophan |
| Gly | glycine |
| RW | raw water |
| FW | finished water |
| SCX | strong cation exchange filler |
| JX, NXQ, QT, XF, CX, and TH | six waterworks in Zehjiang province |

### 3.1. Principle of the Method

With carboxyl groups (-COOH), amine groups (-NH$_2$), and hydroxyl groups (-OH) in their molecular structures, FAAs usually have strong polarity and poor volatility, and therefore cannot be detected directly through GC-MS [22]. Through derivatization, FAAs can be transformed to corresponding less polar derivatives which are suitable for injection and separation by the GC column. Through derivatization by hydrochloric acid/n-butanol and trifluoroacetic anhydride, the -COOH of the FAAs becomes butylated and the -OH and -NH$_2$ are acylated [23]. The reaction mechanism is as follows:

$$RCHNH_2COOH \xrightarrow{n-butanol} RCHNH_2COOC_4H_9 \xrightarrow{trifluoroacetic\ anhydride} RCHNHOCOCF_3COOC_4H_9$$

To increase sensitivity and avoid analysis errors being introduced by impurities produced in the derivatization process, the selected ion mode (SIM) was chosen for quantification after finding these ions in full scanning mode. The qualitative and quantitative analysis of FAAs relies on peak retention time and peak area. Characteristic and interference peaks of the target FAAs are listed in Table 3.

Table 3 shows that the different amino acid derivatives have various characteristic peaks. Each amino acid has a characteristic peak [M-101]$^+$, which represents the molecular ion losing -COOC$_4$H$_9$ [24]. The [M-101]$^+$ ion is the base peak of the Ala, Val, and Pro derivatives. The base peaks of Phe and Glu are m/z 91 and 180, respectively. The base peaks of Thr, Ser, Leu, Ile, and Asp are m/z 57, 57, 69, 69, and 57; these m/z values are low mass fragments for many molecules. Therefore, the next most intense characteristic ions were selected for SIM, which are m/z 153, 139, 182, 182, and 240, respectively, for Thr, Ser, Leu, Ile, and Asp. The use of SIM avoids interferences and improves the sensitivity of the analysis. Figure 2 shows the chromatogram of 10 free amino acids derived by n-butanol and trifluoroacetic anhydride.

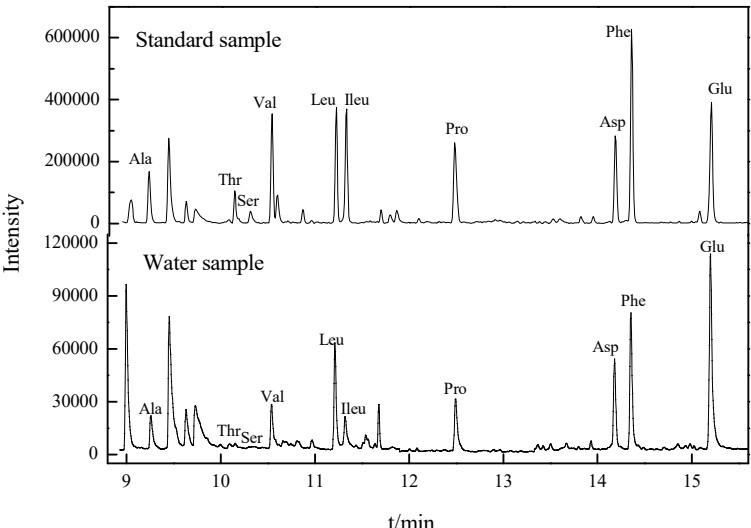

**Figure 2.** Gas chromatography–mass spectrometry (GC-MS) chromatogram of standard free amino acids (FAAs) and raw water sampled from JX waterworks.

### 3.2. Calibration Curves

The FAA stock solution contained 10 amino acids at a concentration of 10 g/L, dissolved in 0.1 M HCl. Mixed FAA standard solutions with concentrations from 0.1 to 30 mg/L were prepared, derivatized, and analyzed. External calibration curves were generated for each FAA and demonstrated good linearity, with correlation coefficients between 0.992 and 0.999. The quantitative detection limits ranged from 2.4 to 13.9 µg/L; individual detection limits for each FAA are presented in Table 3.

**Table 3.** Characteristic ions and retention times of FAA derivatives.

| Free Amino Acid | Derivative Molecular Ion (m/z) | Base Peak (m/z) | Characteristic Ions (m/z) | Retention Time (min) | Limit of Detection (µg/L) |
|---|---|---|---|---|---|
| Ala | 241 | 140 | 41, 57, 69,140 | 9.43 | 5.6 |
| Thr | 271 | 57 | 41, 57, 69, 153 | 10.13 | 10.8 |
| Ser | 353 | 57 | 57, 69, 41, 139 | 10.30 | 13.9 |
| Val | 269 | 168 | 168, 55, 41,57, 168 | 10.53 | 5.1 |
| Leu | 283 | 69 | 41, 182, 140, 57, 69 | 11.21 | 2.9 |
| Ileu | 283 | 69 | 41, 57, 152, 69, 182 | 11.32 | 3.2 |
| Pro | 267 | 166 | 41, 69, 96, 166 | 12.47 | 2.3 |
| Asp | 341 | 57 | 41, 240, 139, 184, 57 | 14.18 | 3.9 |
| Phe | 317 | 91 | 41, 148, 204, 91 | 14.36 | 2.4 |
| Glu | 355 | 180 | 41, 57, 180, 152, 198, 254 | 15.20 | 3.4 |

### 3.3. Reliability Assessment of FAA Derivatization

The reliability of FAA derivatization was investigated by recovery analysis. Concentrations of 5, 10, and 20 µg/L mixed free amino acids from the standard solution were added into pure water, tap water of CX waterworks, finished water of CX waterworks (pH 7.1; TOC 2.7 mg/L; turbidity 0.21 NTU; and conductivity 179 µs/cm), and raw water of CX waterworks drawn from Tiaoxi River (pH 7.5; TOC 9.6 mg/L; turbidity 120 NTU; and conductivity 238 µs/cm). A 1 L sample was enriched by solid phase extraction followed by derivatization and injected into GC-MS for quantitative analysis. The recoveries of various FAAs are illustrated in Table 4.

**Table 4.** SPE-derivatization recovery rates of FAAs in pure water, tap water, and raw water.

| FAA | Pure Water (%) | | | Tap Water (%) | | | Raw Water (%) | | | Mean (s.d) (%) |
|---|---|---|---|---|---|---|---|---|---|---|
| | 5 | 10 | 20 | 5 | 10 | 20 | 5 | 10 | 20 | |
| | (µg/L) | | | (µg/L) | | | (µg/L) | | | |
| Ala | 84.2 ± 1.2 | 82.6 ± 2.3 | 84.3 ± 0.9 | 86.6 ± 4.5 | 87.6 ± 2.5 | 85.4 ± 1.8 | 80.4 ± 3.6 | 82.6 ± 1.3 | 89.4 ± 3.4 | 84.8 (2.8) |
| Thr | 72.3 ± 2.5 | 74.2 ± 1.6 | 77.3 ± 2.3 | 67.6 ± 3.2 | 65.8 ± 3.4 | 65.4 ± 4.6 | 70.3 ± 2.4 | 72.8 ± 2.6 | 69.0 ± 2.9 | 74.5 (4.0) |
| Ser | 89.7 ± 1.9 | 95.6 ± 2.8 | 92.2 ± 1.8 | 71.4 ± 3.8 | 73.9 ± 1.8 | 72.3 ± 5.1 | 63.5 ± 3.1 | 70.9 ± 1.4 | 76.6 ± 1.2 | 78.5 (11.2) |
| Val | 86.5 ± 1.8 | 84.4 ± 3.1 | 87.6 ± 1.7 | 94.3 ± 1.6 | 90.6 ± 1.3 | 92.3 ± 3.8 | 91.3 ± 1.9 | 92.6 ± 3.7 | 90.0 ± 4.7 | 90.0 (3.2) |
| Leu | 83.2 ± 2.7 | 80.4 ± 4.7 | 82.6 ± 3.5 | 82.3 ± 2.2 | 79.0 ± 2.6 | 86.8 ± 4.5 | 80.2 ± 1.6 | 85.6 ± 4.2 | 83.1 ± 5.3 | 82.6 (2.5) |
| Ileu | 85.4 ± 3.6 | 89.6 ± 2.9 | 82.4 ± 4.7 | 87.8 ± 3.4 | 86.6 ± 3.7 | 92.4 ± 3.2 | 87.7 ± 2.8 | 85.6 ± 3.8 | 88.4 ± 2.7 | 87.3 (2.8) |
| Pro | 87.6 ± 5.2 | 85.5 ± 4.1 | 80.4 ± 2.6 | 85.6 ± 1.1 | 87.5 ± 2.9 | 76.5 ± 2.8 | 87.1 ± 3.4 | 89.4 ± 4.9 | 90.4 ± 3.8 | 85.6 (4.4) |
| Asp | 73.5 ± 1.3 | 71.1 ± 3.5 | 75.6 ± 1.5 | 72.3 ± 3.6 | 73.7 ± 1.2 | 81.5 ± 1.6 | 72.3 ± 5.4 | 72.9 ± 4.6 | 76.5 ± 1.6 | 74.4 (3.2) |
| Phe | 89.4 ± 3.4 | 90.2 ± 2.4 | 92.4 ± 1.2 | 79.1 ± 1.5 | 83.8 ± 0.8 | 75.4 ± 1.9 | 83.1 ± 3.2 | 87.8 ± 2.5 | 84.5 ± 2.9 | 85.1 (5.5) |
| Glu | 79.1 ± 4.1 | 78.8 ± 1.3 | 76.8 ± 4.8 | 72.8 ± 2.4 | 73.6 ± 5.3 | 75.8 ± 3.7 | 75.2 ± 1.4 | 73.5 ± 1.9 | 76.7 ± 3.8 | 75.8 (2.3) |
| Mean FAAs | 81.3 | 83.2 | 83.2 | 80.0 | 80.2 | 80.4 | 79.1 | 81.4 | 82.5 | 81.5 (1.6) |
| s.d. FAAs | 6.2 | 7.5 | 6.0 | 8.7 | 8.2 | 8.0 | 8.7 | 8.1 | 7.4 | |

"%" means that percent recovery of ten amino acids desolved in three types of water after SPE-derivatization.

As shown in Table 4, overall recovery for derivatized FAAs was approximately 80%, with similar recoveries under three different water quality conditions and three concentrations of 5, 10, and 20 µg/L. Recoveries for individual FAAs ranged from 63.5% to 95.6%. Recoveries of >80% were observed for Ala, Val, Leu, Ile, Pro, and Phe in all three different water qualities. Overall recoveries of Ser, Thr, Asp, and Glu were about 75%. The recovery of Ser was the most variable; >90% recovery was obtained in distilled water, while recovery was about 20% lower in tap and raw water. The Ser recovery in raw water also varied with concentration. In reference to the detection quality control of other contaminants in water, the above results show that the SPE-derivatization pretreatment method meets the requirements of free amino acid detection [25].

### 3.4. The Occurrence of FAAs in Source and Treated Water Samples

The chromatogram of raw water from JX waterworks, as a typical example, is shown in Figure 2. The water quality of samples is shown in Table 1. Residual chlorine in finished water samples ranged from 1.57 to 2.02 mg/L. Table 5 shows the FAA occurrence in raw water and finished water of water

treatment plants. Ten target standard amino acids in ultrapure water were measured, and most showed high sensitivity. Compared with ultrapure water, the chromatogram for analysis of amino acids in source water showed similar sharp peaks and little interference, indicating that GC/MS coupled with SPE performs well for complicated water samples.

As shown in Figure 3, the average concentration of the ten FAAs in T Lake was 15.3 µg/L, which was less variable than that previously reported by Xin et al. [26]. They found that AA concentrations of T lake were 2.59, 0.48, and 0.48 µmol/L in north, south, and east positions, respectively. Algae acted as the main source of AAs, and their high concentration and seasonal fluctuation lead to varied AAs at a high level in water. The amount of total FAAs in sewage (up to 35.4 µg/L) was higher than that in source water, and was decreased by 72% through biological treatment and the chlorine disinfection process. The average total FAAs in source water, in the range of 20.0–20.3 µg/L, were removed partly by the physical and chemical treatment processes employed in drinking water treatment. Advanced treatment processes in TH waterworks removed 80% of the FAAs, much more than the conventional methods adapted in CX waterworks (39.2%). Similar compositions of FAAs were found in sewage and source water; the most abundant amino acid was Ala in all water types, followed by Ser. Together, Ala and Ser accounted for 30–50% of the total FAAs.

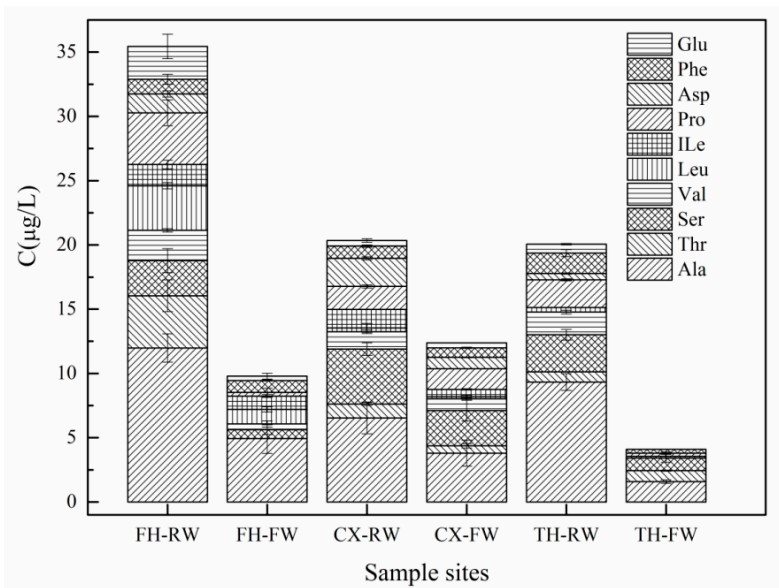

**Figure 3.** Annual average concentration of total FAAs in FH wastewater plant and CX and TH waterworks.

In source waters, the total FAAs for the 10 FAAs measured varied in the range of 6.4–45.9 µg/L; this is a little lower than previous studies that measured more FAAs [12,14,27]. The percent occurrences of most FAAs in source waters were high: Ala 96%, Thr 100%, Ser 92%, Val 96%, Leu 88%, Pro 83%, and Phe 83%, while Ile, Glu, and Asp were below 67%. FAAs occurred in the ranges of ND–14.73 µg/L Ala, 0.43–6.58 µg/L Thr, ND–6.68 µg/L Val, ND–7.84 µg/L Ser, ND–4.16 µg/L Leu, ND–1.70 µg/L Ile, ND–8.06 µg/L Pro, ND–4.73 µg/L Asp, ND–5.19 µg/L Phe, and ND–4.82 µg/L Glu (Table 1). Among these free amino acids, Ala was found in the highest concentrations, followed by Ser, Pro, and Val; Ile had the lowest occurrence and concentration. In previous research, tyrosine, leucine, isoleucine, glysine, alanine, and serine were reported to be the most frequent AAs in natural water [12,14,27,28]. The amount, composition, and dominant species of FAAs in source water may change along with the seasons, as concentrations mainly depend on the variation of biological sources, such as algae [12]. Seasonal sewage discharge and runoff may also influence FAA concentrations.

**Table 5.** The concentrations of various FAAs in drinking water treatment plants in Zhejiang Province.

| Waterworks | | JX Waterworks (µg/L) | | NXQ Waterworks (µg/L) | | QTWaterworks (µg/L) | | XF Waterworks (µg/L) | | CXWaterworks (µg/L) | | THWaterworks (µg/L) | |
|---|---|---|---|---|---|---|---|---|---|---|---|---|---|
| Treatment Process | | Surface Water-Coagulation, Flocculation, Sedimentation, Filtration, Chlorination | | Surface Water-Coagulation, Flocculation, Sedimentation, Ozonation-Bioactive Carbon, Filtration, Chlorination | | Surface Water-Flocculation, Sedimentation, Filtration, Micromembrane, Chlorination | | Surface Water-Coagulation, Flocculation, Sedimentation, Filtration, Chlorination | | Surface Water-Coagulation, Flocculation, Sedimentation, Filtration, Chlorination | | Surface Water-Preozonation, Coagulation, Flocculation, Sedimentation, Ozonation, Active Carbon Filtration, Chlorination | |
| Free Amino Acid | Date | Raw Water | Finished Water | Raw Water | Finished Water | Raw Water | Finished Water | Raw Water | Finished Water | Raw Water | Finished Water | Raw Water | Finished Water |
| Ala | ① | 7.39 | 7.49 | 14.73 | 10.10 | 9.27 | 4.68 | 11.36 | 7.78 | 6.34 | 4.33 | 10.23 | 1.56 |
| | ② | 6.61 | 1.48 | 14.36 | 12.70 | 8.02 | 3.56 | 14.10 | 6.15 | 6.73 | 3.28 | 8.43 | 1.62 |
| | ③ | 6.93 | 1.77 | 5.00 | 2.79 | 8.21 | 1.73 | 4.27 | 2.19 | 0.57 | 0.45 | ND | ND |
| | ④ | 5.81 | – | 4.73 | – | 5.52 | – | – | – | 4.27 | 2.14 | 9.23 | 1.97 |
| Thr | ① | 1.44 | 1.16 | 2.39 | 0.93 | 0.87 | ND | 1.97 | 1.79 | 0.93 | 0.50 | 1.02 | 0.85 |
| | ② | 1.69 | 0.30 | 1.41 | 0.52 | 0.47 | ND | 0.57 | 0.20 | 1.23 | 0.64 | 0.55 | ND |
| | ③ | 0.92 | 0.34 | 0.43 | 0.55 | 0.48 | 1.49 | 0.72 | 0.21 | – | – | – | – |
| | ④ | 1.07 | – | 1.24 | – | 1.30 | – | – | – | 0.48 | 0.12 | 6.58 | 2.01 |
| Ser | ① | 7.84 | 7.26 | ND | ND | 3.81 | 2.15 | ND | ND | 3.45 | 3.01 | 3.71 | 0.98 |
| | ② | 5.19 | 3.83 | 1.78 | 1.12 | 1.21 | 0.41 | 1.67 | 0.52 | 5.09 | 2.47 | 2.06 | ND |
| | ③ | 1.25 | 1.01 | 1.83 | 1.11 | 1.26 | 0.52 | 1.70 | 0.62 | 1.80 | 3.23 | 0.60 | 0.88 |
| | ④ | 5.00 | – | 3.58 | – | 3.94 | – | – | – | 2.46 | 1.25 | 2.38 | 0.96 |
| Val | ① | 3.19 | 1.40 | 6.86 | 1.64 | ND | ND | 6.64 | 2.00 | 1.31 | 0.71 | 1.46 | ND |
| | ② | 3.23 | 1.24 | 5.97 | 2.36 | 0.40 | 0.23 | 6.20 | 3.14 | 1.39 | 0.21 | 0.55 | ND |
| | ③ | 0.24 | 0.23 | 0.74 | 0.56 | 0.50 | 0.22 | 0.63 | ND | 0.99 | 0.85 | 0.49 | 0.24 |
| | ④ | 1.77 | – | 1.64 | – | 1.52 | – | – | – | 2.04 | 0.98 | 1.13 | 0.12 |
| Leu | ① | 2.20 | 1.80 | 3.07 | 1.89 | 3.00 | 1.90 | 4.10 | 2.02 | 0.13 | 0.11 | 0.17 | 0.14 |
| | ② | 3.48 | 1.56 | 4.16 | 1.29 | 4.01 | 1.72 | 3.90 | 2.40 | 0.47 | 0.12 | ND | ND |
| | ③ | 0.56 | 0.37 | 0.55 | 0.48 | 0.43 | 0.18 | 0.34 | 0.47 | ND | ND | ND | ND |
| | ④ | 1.77 | – | 1.38 | – | 2.13 | – | – | – | 0.19 | ND | 0.12 | ND |
| Ile | ① | ND | ND | ND | ND | ND | ND | ND | ND | 1.54 | 0.57 | ND | ND |
| | ② | ND | ND | ND | ND | ND | ND | ND | ND | 1.34 | ND | ND | ND |
| | ③ | 1.54 | ND | 0.05 | 0.02 | ND | ND | 0.08 | ND | ND | 0.47 | 0.21 | 0.62 |
| | ④ | 1.08 | – | 1.65 | – | 1.70 | – | – | – | ND | ND | ND | ND |

**Table 5.** *Cont.*

| Waterworks | | JX Waterworks (µg/L) | | NXQ Waterworks (µg/L) | | QTWaterworks (µg/L) | | XF Waterworks (µg/L) | | CXWaterworks (µg/L) | | THWaterworks (µg/L) | |
|---|---|---|---|---|---|---|---|---|---|---|---|---|---|
| Treatment Process | | Surface Water-Coagulation, Flocculation, Sedimentation, Filtration, Chlorination | | Surface Water-Coagulation, Flocculation, Sedimentation, Ozonation-Bioactive Carbon, Filtration, Chlorination | | Surface Water-Flocculation, Sedimentation, Filtration, Micromembrane, Chlorination | | Surface Water-Coagulation, Flocculation, Sedimentation, Filtration, Chlorination | | Surface Water-Coagulation, Flocculation, Sedimentation, Filtration, Chlorination | | Surface Water-Preozonation, Coagulation, Flocculation, Sedimentation, Ozonation, Active Carbon Filtration, Chlorination | |
| Free Amino Acid | Date | Raw Water | Finished Water | Raw Water | Finished Water | Raw Water | Finished Water | Raw Water | Finished Water | Raw Water | Finished Water | Raw Water | Finished Water |
| Pro | ① | ND | ND | 8.06 | 1.80 | 1.90 | 0.84 | 4.15 | ND | 1.80 | 1.60 | 2.48 | ND |
| | ② | 1.20 | 0.67 | 6.19 | 0.96 | 0.98 | 0.64 | 1.02 | 0.60 | ND | ND | 1.84 | 0.24 |
| | ③ | 1.39 | 0.77 | 1.30 | 1.01 | 1.05 | 0.75 | 1.03 | 0.63 | ND | ND | ND | ND |
| | ④ | 3.03 | – | 2.85 | – | 3.54 | – | – | – | 2.41 | 1.75 | 3.09 | 1.04 |
| Asp | ① | 2.11 | 1.24 | 4.73 | 1.39 | 2.12 | ND | 3.42 | 1.91 | ND | 0.90 | ND | ND |
| | ② | 2.52 | 0.69 | 2.91 | 0.74 | 0.65 | 0.42 | 0.94 | ND | 2.18 | 0.88 | ND | 0.49 |
| | ③ | 3.92 | 0.80 | 1.62 | 0.89 | 0.69 | 0.49 | 1.02 | ND | ND | ND | ND | ND |
| | ④ | ND | – | ND | – | ND | – | – | – | 0.62 | ND | 0.27 | ND |
| Phe | ① | 1.13 | ND | 2.26 | 1.28 | 1.22 | 0.44 | 1.76 | 1.16 | 1.00 | 0.41 | 1.59 | ND |
| | ② | 3.01 | ND | 1.59 | ND | 0.46 | 0.31 | 1.59 | 0.41 | 0.89 | 1.07 | ND | 0.30 |
| | ③ | 1.81 | ND | 1.45 | 0.59 | ND | 0.69 | ND | ND | 5.19 | 5.19 | 5.14 | 4.27 |
| | ④ | ND | – | 1.50 | – | 1.58 | – | – | – | 1.37 | 0.78 | 1.63 | 0.74 |
| Glu | ① | ND | ND | 3.83 | 0.72 | 1.00 | 0.29 | 2.00 | 1.43 | ND | 0.66 | ND | 0.69 |
| | ② | 0.41 | 0.23 | 2.20 | 0.85 | 1.30 | ND | 3.41 | ND | 0.44 | 0.12 | ND | ND |
| | ③ | 4.82 | 0.42 | 4.59 | 0.23 | ND | 0.08 | ND | 0.13 | ND | ND | ND | ND |
| | ④ | ND | – | ND | – | ND | – | – | – | 0.51 | 0.12 | ND | ND |
| Total FAAs | ① | 25.30 | 20.35 | 45.93 | 19.75 | 23.19 | 10.30 | 35.40 | 18.09 | 16.50 | 12.80 | 20.66 | 4.22 |
| | ② | 27.34 | 10.00 | 40.57 | 20.54 | 17.50 | 7.29 | 33.40 | 13.42 | 19.76 | 8.79 | 13.92 | 2.65 |
| | ③ | 23.38 | 5.71 | 17.56 | 8.23 | 12.62 | 6.15 | 9.79 | 4.25 | 8.55 | 10.19 | 6.44 | 6.01 |
| | ④ | 19.53 | – | 18.57 | – | 21.23 | – | – | – | 14.35 | 7.14 | 24.43 | 6.84 |

### 3.5. The Degradation of FAAs in Different Water Treatment Plants

FAA concentrations in finished water were always lower than in the corresponding raw water. The total FAA concentrations in finished water were 16–78% lower than in raw water; removal occurred by ozonation, chlorination, and transformation to DBPs. FAAs were measured in water samples collected from each unit of the water treatment process in CX waterworks and TH waterworks. The total FAA occurrence after individual conventional and advanced treatment processes is shown in Figure 4.

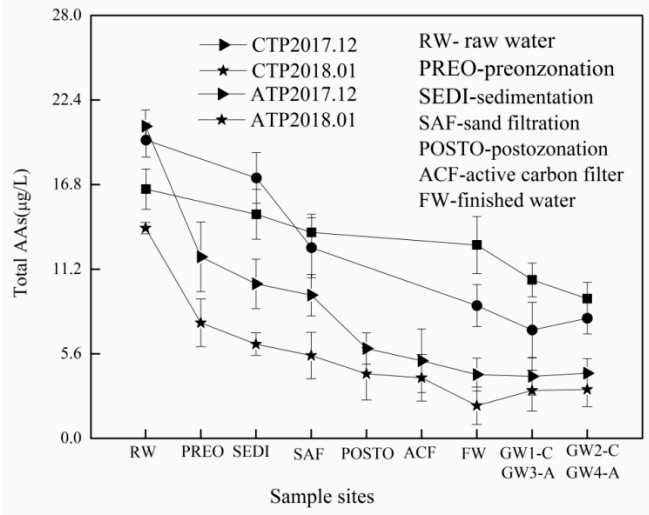

**Figure 4.** The variation of total FAAs in conventional and advanced treatment processes. (CTP is CX).

As can be seen, preozonation, coagulation sedimentation, sand filtration, post ozonation, activated carbon filtration, and chlorination each remove a portion of the FAAs. The removal efficiency of conventional treatment processes was lower than that of advanced treatment processes. Ozone oxidation had a strong effect on FAA removal, preozonation removal efficiency was 45%, and postozonation contributed an additional 20% removal. Ozonation can alter the molecular structures of the precursors; subsequently, these precursors were adsorbed by activated carbon. Ozonation coupled with biological activated carbon ($O_3$-BAC) can significantly reduce the precursors of regulated carbon DBPs(C-DBPs) and trace organic contaminants [29]. The decrease of the FAA concentration in finished water compared with that in samples with an activated carbon filter indicated that chlorine can react with FAAs in the disinfection process. Trihalomethanes and haloacetic acids were prevalent carbonaceous DBPs derived from FAAs. Additionally, nitrogenous DBPs can also be derived from FAAs.

### 3.6. DBP Formation Potential of Typical FAAs

Phe, Glu, and Asp, three amino acids which occurred in the source water of H city, together with Tyrosine (Tyr), were chlorinated and examined for the formation potential of DBPs; the results are shown in Figure 5. Table 6 summarizes studies that demonstrate the amino acids are precursors to DBPs. Multiple THMs, HAAs, HANs, and TCNM were measured; only TCM, dichloroacetic acid (DCAA), trichloroacetic acid (TCAA), dichloroacetonitrile (DCAN), and trichloronitromethane (TCNM) were detected in the samples.

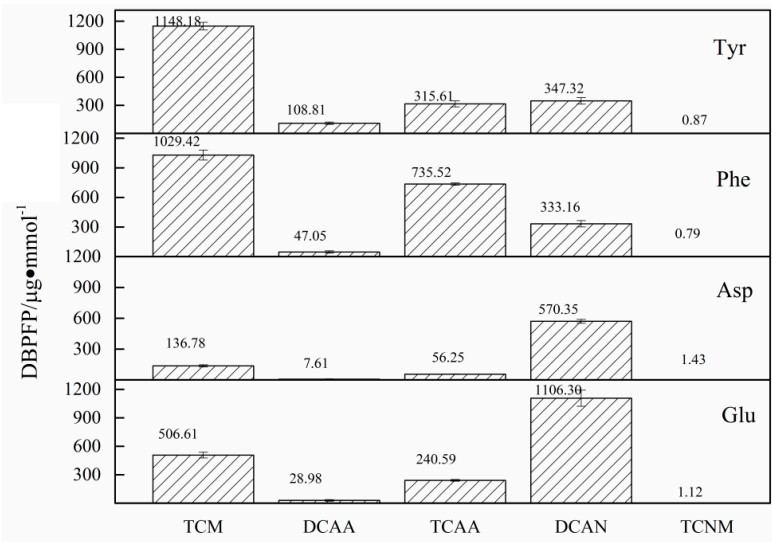

**Figure 5.** The disinfection byproduct formation potential of Tyr, Phe, Glu, and Asp during chlorination. TCM = trichloromethane; DCAA = dichloroacetic acid; TCAA = trichloroacetic acid; DCAN = dichloroacetonitrile; TCMN = trichloronitromethane.

**Table 6.** FAAs and corresponding disinfection byproducts (DBPs).

| FAA | Disinfectant | DBPs | Reference |
| --- | --- | --- | --- |
| Ala | Chlorination | MCAld, DCAld, TCAld, CF | [5] |
| Thr | Chlorination | TCAce | [30] |
| Leu | Chlorination | CF | [31] |
| Asp | Chlorination | CH, DCAN, TCNM, DcAcAm, CF | [3] [31] |
| Phe | Chlorination | Phenylacetonitrile, phenylacetaldehyde, benzyl chloride, 2-chlorobenzyl cyanide, 2, 6-diphenylpyridine | [32] |
| Glu | Chlorination | CF | [31] |
| Tyr | Chlorination | CH, DCAN, TCAN, DCAA, TCAA, TCNM, CNCl,4-CP | [23] [3,23] [3] |
| Gly | Chlorination | THMs, CF, DCAN, CNCl | [33] [3] |
| Trp | Chlorination | CH, DCAN | [3] |
| His | Chlorination | CF, DCAN, DCAcAm | [33] |
| Arg | Chlorination | CF | [31] |

MCAld represents monochloroacetaldehyde acetaldehyde; DCAld represents dichloroacetaldehyde; TCAld represents trichloroacetaldehyde; CF represents chloroform; TCAce represents trichloroacetone; CH represents chloralhydrate; TCNM represents trichloronitromethane; DcAcAm represents dichloroacetamide; CNCl represents cyanogen chloride; 4-CP represents 4-chlorophenol.

The DBP formation potential results indicate that amino acids with similar structures may show the same characteristics in generating DBPs. Phe and Tyr are amino acids possessing an aromatic ring; Phe possesses a phenyl functional group, and Tyr possesses a p-hydroxyphenyl group. Phe and Tyr exhibited similar patterns, with higher TCM yields than DCAN. The DCAA yield of Phe was higher than Tyr, which was not in accordance with previous reports [28]; however, the Cl demand of Tyr (14.8 mol/molAA) was higher than Phe (3.6 mol/molAA). Phe and Tyr were reported to produce more TCAA than DCAA, since they contained more conjugated systems. Asp and Glu, with chain structures and a low C:N ratio, produced a higher amount of N-DBPs than C-DBPs. Chu et al. found that the cytotoxicity and genotoxicity of DCAN and TCNM were much higher than HAAs; 50% of cells were killed when the concentrations of DCAN and TCNM were only $10^{-4}$ and $10^{-3}$, respectively [34]. Overall, N-DBPs produced by amino acids cannot be ignored during chlorine disinfection.

## 4. Conclusions

In H City, Zhejiang, the source of drinking water has recently been supplied by long-distance transportation. But it is not a permanent fix. In the water transfer project, sewage treatment (i.e., in rural areas) should be treated to reduce the pollution of surface waters, especially water sources pollution. The results of this study revealed that free amino acids (FAAs) existed in all treatment units of drinking water, and the concentrations of FAAs varied and obviously decreased in chlorine disinfection. Amino acids are chosen as the model precursors of emerging nitrogen-containing disinfection by-products (N-DBPs); hence, it is possible to generate various DBPs during chlorination, and its concentration should be strictly under control.

SPE and derivatization, coupled with GC-MS, was demonstrated as an effective and reliable method for the identification and quantification of low μg/L concentrations of free amino acids in raw and treated water. Through SPE-GC-MS, the FAA occurrence in the raw water and finished water of six drinking waterworks in Zhejiang province was confirmed. In source raw waters, FAAs were found in the ranges of 1.48–14.73 μg/L Ala, 0.20–2.39 μg/L Thr, 0.41–7.84 μg/L Val, 0.21–6.86 μg/L Ser, 0.11–4.16 μg/L Leu, 0.57–1.54 μg/L Ile, 0.24–8.06 μg/L Pro, 0.42–4.73 μg/L Asp, 0.30–3.01 μg/L Phe, and 0.12–3.83 μg/L Glu. FAA concentrations were always lower in finished drinking water, indicating removal by treatment or transformation to C-DBPs and N-DBPs. Conventional water treatment was demonstrated to be less effective than ozonation or activated carbon in removing FAAs. When chlorinated, Phe, Tyr, Asp, and Glu produced trichloromethane, dichloroacetic acid, trichloroacetic acid, dichloroacetonitrile, and trichloronitromethane, demonstrating that FAAs are precursors for both C-DBPs and N-DBPs. Due to the suspected higher toxicity of many N-DBPs compared to C-DBPs, there is an urgent need to investigate the occurrence of FAAs in source and drinking water systems. Additionally, other factors, such as pipe material and microorganisms in the pipe, can affect the concentration of DBPs and need to be further analyzed.

**Author Contributions:** Conceptualization, X.M., Q.Y.; investigation, Q.Y., J.F.; data curation, X.L., K.Z.; writing—original draft preparation, Y.Y., R.Z.; writing—review and editing, Y.Y., R.Z.; supervision, A.M.D.; funding acquisition, X.M. All authors have read and agreed to the published version of the manuscript.

**Funding:** The authors appreciate the funding support from Major Science and Technology Program for Water Pollution and Treatment (2017ZX07201004), National Natural Science Foundation of China (Grant No. 51678527, 51208468 and No. 51878582), and the Natural Science Foundation of Zhejiang Province of China (Grant No. LY19E080019).

**Conflicts of Interest:** The authors declare no conflict of interest.

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
