# Peer review of "Occurrence of Free Amino Acids in the Source Waters of Zhejiang Province, China, and Their Removal and Transformation in Drinking Water Systems"

_water, doi:10.3390/w12010073_

Round 1

Reviewer 1 Report

Severe critics adressing the authors: 1) The English is of very low quality, concerning grammar as well as syntax, - astonishing, since one of the authors (AD)seems to be a native English Speaker! Some acronyms used in the text, should have been explained in a glossary, example the term "DBP", - the reader not so well familiar with the topic is expected to guess, that it stands for "disinfection byproducts". 2) The detailed description of the analytical work (chapters 2.2. - 2.3.) seems to be rather unnecessary, because it's well described in literature elsewhere over many years, could be defined as "standard procedure ". 3) Structure, logic of the paper presented:

Some comment about the source of the FAA (free amino acids) in the raw water samples is missing: Are they the left-overs of urban/industrial sewage waters in the investigated samples or are they taken up from the soil in the corresponding ground water. An existence of FAAs seem to hint to a large load of decomposed proteinaceous carbon in the water?

Throughout the text, tables, and figures the term sewage, raw, and source water is used confusingly  for the same water samples as compared to finished waters. The reader would be keen to learn about the sources of the unfinished water samples drawn. Were it possible that, due to the immense  urban population of Zhejiang, drinking water must be produced directly from sewage water?

The authors really report - documented by their detailed results - that the conventional water treatment really results in a production of unwanted and even toxic(!) byproducts in the finished drinking waters, quite in contrast to the desired "removal" of contaminants. It would have been interesting to quantify the "efficiency" in terms of percentage of how much DBPs were produced from the educts FAAs. This extremely important result of the paper is not worked out explicitly in the conclusion at all! The overall warning should definitely be more expressed in the end of the text. A possible, -perhaps the only solution - to the problem would be a strict separation of sewage/waste water from potential raw water to be treated to serve as drinking water  - taken from gound or surface elsewhere in sufficient distance from sewage/waste effluent!

I hope that the results of this paper would be made public in order to induce a radical improvement of drinking water supply in Zhejiang, but the whole protocoll of this truly important work should be also radically improved to meet the ambitious aim of the authors.  

Everything is said above in my comment to the authors. I definitely do advocate the publication of the results, but not in the present form, which is deplorable, both from the language and the structure plus conclusion. (It's possible that the authors do not want to take the risk of being more pronounced to worry about the quality of conventional water treatment in their home province Zhejiang?) But I wonder why the coauthor AD from USA did not help to overcome the obvious problems? Language- and content-wise...Publishing in China these days is not easy, I guess. I suggest the editor should give some hints how to help the authors to send their message - supported by their great efforts -  to the public.   

Author Response

Dear Reviewer,

On behalf of our coauthors, I would like to express our sincere thanks for giving this opportunity to revise our manuscript. We appreciate your critical evaluation, and meaningful comments and suggestions on our manuscript. All comments helped us to improve the quality of our manuscript. According to your comments and Editor’s direction, we have carefully revised our manuscript for possible consideration of publication. 

We have taken special care to revise the entire manuscript by rewording and rephrasing. For the best quality of English, the revised version of manuscript has been edited by MDPI language services (english-14701) and the Certificate is attached below.

In addition, we considered your appreciations, advises, and comments regarding Originality, Significance, Quality of presentation, Scientific soundness, Interest to readers, and Overall merit of our paper. Wherever necessary, we have provided additional supporting information according to the comments. All corrections in the revised manuscript were marked in red color.  Our detailed ‘point-by-point’ response to each of your comment has been provided below. We do believe that the revised manuscript is more suitable for your valuable decision.

Comment 1:The detailed description of the analytical work (chapters 2.2. - 2.3.) seems to be rather unnecessary, because it's well described in literature elsewhere over many years, could be defined as "standard procedure ". 

Response: As suggested, the chapters have been supplemented in the manuscript, Page 3-4.  Though there were literatures reported the analytical work,  but detection method for micro level amino acids were not thoroughly introduces.

Comment 2:Some comment about the source of the FAA (free amino acids) in the raw water samples is missing: Are they the left-overs of urban/industrial sewage waters in the investigated samples or are they taken up from the soil in the corresponding ground water. An existence of FAAs seem to hint to a large load of decomposed proteinaceous carbon in the water?

Response: we have supplemented information in the revised manuscript, Page 4, Lines 153-154. The source water was taken from river ,  which also receiveedurban sewage discharged  upstream far away from the intake. Samples were draw from a target sewage water treatment plant, since nitrogen-containing organic matter in domestic sewage is one of the sources of amino acid. On the other hand, rivers receive pollutants from farmland and forest land in the river basin by rainwash, and contaminants from dry or wet atmospheric deposition. Moreover, algae in rivers can also induce amino acids through metabolism. Except domestic sewage water, other sources mentioned above were not investigated.

Comment 3:Throughout the text, tables, and figures the term sewage, raw, and source water is used confusingly  for the same water samples as compared to finished waters. The reader would be keen to learn about the sources of the unfinished water samples drawn. Were it possible that, due to the immense  urban population of Zhejiang, drinking water must be produced directly from sewage water?

Response: We are apologizing for the confusing name of water samples. The  explanation of pure, raw, source, tap and finished water are shown in text(Page 5, Lines 193-196). Pure water is generated from a Milli-Q water purification system in the lab. Raw waterrepresent source water of waterworks; tap water is taken from the distribution network; finished water iseffluent of  waterwork.

We would like to bring to Reviewer’s kind notice thatwe didn't mean to reclaim the sewage water as water source.We collected samples from sewage plants, surface water, effluent from treatment process of drinking water treatment plants, and water supply networks. We try to analyze the amino acids occurrence in the whole social watercycle. the discharge of the target sewage plant into the nearby waters, which is far away from the water intake position of the water plant.The purpose of these data is to provide information about amino acids discharged into the river by the sewage plant.

Comment 4:The authors really report - documented by their detailed results - that the conventional water treatment really results in a production of unwanted and even toxic(!) byproducts in the finished drinking waters, quite in contrast to the desired "removal" of contaminants. It would have been interesting to quantify the "efficiency" in terms of percentage of how much DBPs were produced from the educts FAAs. This extremely important result of the paper is not worked out explicitly in the conclusion at all! The overall warning should definitely be more expressed in the end of the text. A possible, -perhaps the only solution - to the problem would be a strict separation of sewage/waste water from potential raw water to be treated to serve as drinking water  - taken from gound or surface elsewhere in sufficient distance from sewage/waste effluent!

Response: The results of this study revealed that free amino acids existed in all treatment units of drinking water, and the concentration of free amino acids(FAA) varied, obvious decreased in chlorine disinfection. Amino acids are chosen as the model precursors of emerging nitrogen-containing disinfection by-products(N-DBPs), so it is possible to generate various DBPs during chlorination. However, not all the amino acids reacting with chlorine were analyzed , four representative amino acids were selected to explain their maximum DBPs formation potential. We admit the reviewer's suggestion is reasonable. Water pollution and toxic tendency of pollutants in water treatment are unavoidable problems in China's water supply industry. For City H, cleaner water are delivered through a long pipe to help solve problem caused by lots of kinds of contaminants. . But it is not a permanent fix. Today  more attention are put on sewage treatment(i.e. in rural areas) , the standard for discharge were enhanced to reduce the pollution of surface waters, especially water sources pollution.

Comment 5:I hope that the results of this paper would be made public in order to induce a radical improvement of drinking water supply in Zhejiang, but the whole protocoll of this truly important work should be also radically improved to meet the ambitious aim of the authors.

Response:In H City, Zhejiang, the source of drinking water has recently been supplied by long-distance transportation. But it is not a permanent fix. In the water transfer project, sewage treatment(i.e. in rural areas) should be treated to reduce the pollution of surface waters, especially water sources pollution.

English edited Certificate from MDPI was shown in document

Reviewer 2 Report

The article “Occurrence of free amino acids in source waters of Zhejiang Province, China and their removal and transformation in drinking water systems” by Yang et. al. is suitable for publication in “Water” after addressing the following minor comments:

1. Page 2, Line 73: “The purpose of this work is to determine occurrence information of FAAs by using a developed solid-phase extraction and derivatization combined with gas chromatography-mass spectrometry (GC-MS) and DBP formation of typical amino acids occurred in water source” Authors should comment on how sensitive these detection methods are compared to other methods such as capillary electrophoresis, near-infrared spectroscopy

2. Page 4, Line 135: Authors should write the reagents on the arrow

3. Authors should also include any reported toxicity parameters/lethal dose associated with N-DBPs

Author Response

Dear Reviewer,

On behalf of our coauthors, I would like to express our sincere thanks for giving this opportunity to revise our manuscript. We appreciate your critical evaluation, and meaningful comments and suggestions on our manuscript. All comments helped us to improve the quality of our manuscript. According to your comments and Editor’s direction, we have carefully revised our manuscript for possible consideration of publication. 

We have taken special care to revise the entire manuscript by rewording and rephrasing. For the best quality of English, the revised version of manuscript has been edited by MDPI language services (english-14701) and the Certificate is attached below. 

In addition, we considered your appreciations, advises, and comments regarding Originality, Significance, Quality of presentation, Scientific soundness, Interest to readers, and Overall merit of our paper. Wherever necessary, we have provided additional supporting information according to the comments. All corrections in the revised manuscript were marked in red color.  Our detailed ‘point-by-point’ response to each of your comment has been provided below. We do believe that the revised manuscript is more suitable for your valuable decision.

Comment 1: Page 2, Line 73: “The purpose of this work is to determine occurrence information of FAAs by using a developed solid-phase extraction and derivatization combined with gas chromatography-mass spectrometry (GC-MS) and DBP formation of typical amino acids occurred in water source” Authors should comment on how sensitive these detection methods are compared to other methods such as capillary electrophoresis, near-infrared spectroscopy

Response: As suggested, the comment on capillary electrophoresis and near-infrared spectroscopy have been supplemented in the manuscript, Page 2, Lines 74-80.

Comment 2:Page 4, Line 135: Authors should write the reagents on the arrow.

Response: n-butanol and trifluoroacetic anhydride as derivatization reagent  were shown on the arrow. (Page 4, Line 170).

Comment 3:  Authors should also include any reported toxicity parameters/lethal dose associated with N-DBPs.

Response:  the comment on toxicity parameters has been supplemented in the manuscript, Page 7, Lines 273-275.

English edited Certificate from MDPI was shown in document.

Reviewer 3 Report

introduction
Line 34:
change "source water" with "water resources" 2. Methods
2.X Add sub-chapter on a description of water sources of Zhejiang Province, China. From where is water? Where and at what time were
sampled. Water quality. Were sampled before treatment or after treatment at waterworks. Please be very clear. It has to be repeatable.
on figure 1 you present raw and standard water sample. Explain what is standard and what raw, pure, tap or finished water sample? Be consistent in using expressions.
Add map with locations.
2.X Add sub-chapter on shortcomings and uncertainties of the study. What could be done better but it was not possible due to various
factors.
2.X add sub-chapter on analysis method used to obtain results (e.g. recovery analysis, ..) Line 127 - General observation
3. Results should on general contain only results. Everything that explains methodology should be moved to 2. Methods.
Please, discuss results in a way that you state if the results are in an acceptable range of they exceed target values or health
standards.
I would expect more discussion about the proposal for future research or proposals to policymakers about regulations. Line 128: This subchapter is trivial should be moved to Methods. From where is this water and why exactly this waterworks was selected to
be present. Line 142
The derivates names (Ala, Val, ...) should be explained somewhere (table, figure, text) Line 164-172
Are these results good or bad? What do they tell us? Line 174-181
This should be in Methods. Make separate sub-chapter 2.X. Line 247
Are these values good or bad? Line 242 - conclusions
Please modify text in a way that you answer to these questions:
1. Why is this research unique?
2. What are the shortcomings/uncertainties?
3. What did the scientific community learn?
4. Benefits for consumers/water managers)?
5. Policy recommendation?
6. Future work? Figures and tables: They need to be self-explanatory! They should stand alone and tell the story. Titles should not contain abbreviations (FAA, DBP, SPE). For example "FAA" should be "Free amino acids (FAA)". Be consistent with all figures and tables. Table 2: are these results for all samples or average of samples form different waterworks. If results come from specific waterwork it should be mentioned.

Author Response

Dear Reviewer,

On behalf of our coauthors, I would like to express our sincere thanks for giving this opportunity to revise our manuscript. We appreciate your critical evaluation, and meaningful comments and suggestions on our manuscript. All comments helped us to improve the quality of our manuscript. According to your comments and Editor’s direction, we have carefully revised our manuscript for possible consideration of publication. 

We have taken special care to revise the entire manuscript by rewording and rephrasing. For the best quality of English, the revised version of manuscript has been edited by MDPI language services (english-14701) and the Certificate is attached below. 

In addition, we considered your appreciations, advises, and comments regarding Originality, Significance, Quality of presentation, Scientific soundness, Interest to readers, and Overall merit of our paper. Wherever necessary, we have provided additional supporting information according to the comments. All corrections in the revised manuscript were marked in red color.  Our detailed ‘point-by-point’ response to each of your comment has been provided below. We do believe that the revised manuscript is more suitable for your valuable decision.

Comment 1:Line 34:change "source water" with "water resources" 2. Methods2.X Add sub-chapter on a description of water sources of Zhejiang Province, China. From where is water? Where and at what time weresampled. Water quality. Were sampled before treatment or after treatment at waterworks. Please be very clear. It has to be repeatable.on figure 1 you present raw and standard water sample. Explain what is standard and what raw, pure, tap or finished water sample? Be consistent in using expressions.Add map with locations

Response: the  "source water" has been revised in the manuscript(Page 1, Line 34).And Sample information(i.e. Water quality, description of water sources and map with locations)has been supplemented in the manuscript(figure 5 and table 4). Morever, The  explanation of pure, raw, source, tap and finished water are shown in text(Page 5, Lines 193-196).Pure water is generated by a Milli-Q water purification system in our lab. Raw water=source water of waterworks; tap water is outlet water at the end of the pipe network; finished water is outlet water at the end of waterwork.

Comment 2:2.X Add sub-chapter on shortcomings and uncertainties of the study. What could be done better but it was not possible due to variousfactors.

Response: According to the comment, shortcomings and uncertainties of the study were added in Page 3, Lines 139-141.

Comment 3:2.X add sub-chapter on analysis method used to obtain results (e.g. recovery analysis, ..)

Response: The results(i.e. linear regression equation and recovery efficiency) of this analysis method were added in Page 3, Lines 114-116.

Comment 4:Results should on general contain only results. Everything that explains methodology should be moved to 2. Methods.
Please, discuss results in a way that you state if the results are in an acceptable range of they exceed target values or healthstandards.

Response:Methodology have been moved to 2. Methods. However, as a small molecule of natural organic matter containing nitrogen, amino acid can produce DBPs in the process of drinking water treatment. As a precursor, amino acid is similar to TOC and can only reflect the pollution situation and the content of precursor. It fails to  get the target values or healthstandards.

Comment 5:I would expect more discussion about the proposal for future research or proposals to policymakers about regulations.

Response: We will consider this research direction in our future work.

Comment 6:The derivates names (Ala, Val, ...) should be explained somewhere (table, figure, text)

Response: We have explainedthe derivates names (Ala, Val, ...) in the text. (Page 2, Lines 59-61).

Comment 7:Line 164-172. Are these results good or bad? What do they tell us?

Response: The supplemented information was shown in the text Page 5, Lines216-220.

Comment 8:Line 174-181. This should be in Methods.

Response: This part has been moved to Methods part.( Page 4, Lines 151-160.)

English edited Certificate from MDPI was shown in document.

Round 2

Reviewer 1 Report

The revised manuscript was improved to a large extent, yet the following points need do be corrected/added: 1) Absolutely necessary is the additon of a Glossary explaining the numerous acronyms in the text, most of them explained in the text/abstract, but some of them open to the guess of the reader! A separate Glossary is a MUST before final acceptance! 2) Still, a thorough check of the English is needed, examples: line 41 "haloaceticacids" (instead of "halosceticacids"), line 219 (grammar) "...and their high...", line 55: For citation of "Berne" no corresponding literature reference found. Do you quote perhaps (12) How et al.?, line 217 (fig. 2): What do you mean by "less variable"? And "T-Lake" the same as "TH-Lake"?, Line 216-220: "respectively" means "after treatment" perhaps? - serve only as a few examples for a rather careless English check! 3) Now in the revised paper there appear 2 citations in Chinese characters, (31) and (27). WATER is an English journal, it is unfair to quote here 2 references in Chinese without English transscript, how should an English reader ever go into the literature to study the original detail quoted? 4) Chuh et al. (30) is cited giving concentrations in terms of "e-4 and e-3", at least unusual! 5) The informative response to comments 4 and 5 is not documented in your final conclusion; why not? If these points were considered, acceptance is recommended.

Author Response

Dear Reviewer,

On behalf of our coauthors, I would like to express our sincere thanks for giving this opportunity to revise our manuscript. We appreciate your critical evaluation, and meaningful comments and suggestions on our manuscript. All comments helped us to improve the quality of our manuscript. According to your comments and Editor’s direction, we have carefully revised our manuscript for possible consideration of publication. 

We have taken special care to revise the entire manuscript by rewording and rephrasing. In addition, we considered your appreciations, advises, and comments regarding Originality, Significance, Quality of presentation, Scientific soundness, Interest to readers, and Overall merit of our paper. Wherever necessary, we have provided additional supporting information according to the comments. All corrections in the revised manuscript were marked in red color.  Our detailed ‘point-by-point’ response to each of your comment has been provided below. We do believe that the revised manuscript is more suitable for your valuable decision.

Comment 1: Absolutely necessary is the additon of a Glossary explaining the numerous acronyms in the text, most of them explained in the text/abstract, but some of them open to the guess of the reader! A separate Glossary is a MUST before final acceptance! 

Response: We really appreciate the reviewer’s advice. We have supplemented a table for explaining the Glossary(Page 4, Line 163-164).

Comment 2: Still, a thorough check of the English is needed, examples: line 41 "haloaceticacids" (instead of "halosceticacids"), line 219 (grammar) "...and their high...", line 55: For citation of "Berne" no corresponding literature reference found. Do you quote perhaps (12) How et al.?, line 217 (fig. 2): What do you mean by "less variable"? And "T-Lake" the same as "TH-Lake"?, Line 216-220: "respectively" means "after treatment" perhaps? - serve only as a few examples for a rather careless English check!

Response: we have revised the entire manuscript(Line 41, 221, 55). We are apologizing for the confusing name of T lake. "T-Lake" is same as "TH-Lake", and we have unified expression for T lake. In addition, “respectively” means different concentration of north, south and east positions in T lake.

Comment 3: Now in the revised paper there appear 2 citations in Chinese characters, (31) and (27). WATER is an English journal, it is unfair to quote here 2 references in Chinese without English transscript, how should an English reader ever go into the literature to study the original detail quoted? 

Response: We are apologizing for the references in Chinese without English transscript. We have corrected the mistake.

Comment 4: Chuh et al. (30) is cited giving concentrations in terms of "e-4 and e-3", at least unusual! 

Response: In Chu et al. study, they used % C1/ 2 to represent the concentration of DBP(disinfection by-product) when 50% of cells were killed. - lg(% C1/ 2)  expressed a positive correlation with toxicity. While 50% of cells were killed, - lg(% C1/ 2) of DCAN and TCNM were 4 and 3, repectively. And thus, the concentration of DCAN and TCNM were 10-4 and 10-3. We have revised it in manuscript(Lines 277-278).

Comment 5:  The informative response to comments 4 and 5 is not documented in your final conclusion; why not? 

Response: We are sincerely thankful for reviewer’s advice. We accept it and add the response to comments 4 and 5 in our final conclusin(Lines 281-288).
